# Study on the Correlations between Quality Indicators of Dry-Aged Beef and Microbial Succession during Fermentation

**DOI:** 10.3390/foods13101552

**Published:** 2024-05-16

**Authors:** Yuliang Cheng, Yiyun Meng, Lin Xu, Hang Yu, Yahui Guo, Yunfei Xie, Weirong Yao, He Qian

**Affiliations:** 1State Key Laboratory of Food Science and Resources, School of Food Science and Technology, Jiangnan University, Wuxi 214122, China; 6220111208@stu.jiangnan.edu.cn (Y.M.); 6190111095@stu.jiangnan.edu.cn (L.X.); yaoweirongcn@jiangnan.edu.cn (W.Y.); qianhe@jiangnan.edu.cn (H.Q.); 2Collaborative Innovation Center of Food Safety and Quality Control in Jiangsu Province, Jiangnan University, Wuxi 214122, China; hangyu@jiangnan.edu.cn (H.Y.); guoyahui@jiangnan.edu.cn (Y.G.); xieyunfei@jiangnan.edu.cn (Y.X.)

**Keywords:** dry aging, beef, microbial succession, flavor, texture

## Abstract

Dry-aged beef has been long favored by people due to its unique flavor and taste. However, the inner relationship between its overall quality formation and microbial changes during dry aging has not yet received much attention and research. To deeply reveal the forming mechanism of the unique flavor and taste of dry-aged beef, correlations between its three main quality indicators, i.e., texture, free amino acids (FAAs), volatile flavor compounds (VFCs), and microbial succession were analyzed in this study. The results showed that *Staphylococcus* spp. and *Macrococcus* spp. were key strains that influenced the total quality of dry-aged beef and strongly correlated with chewiness, hardness, and sweet FAAs (Ala), providing beef with unique palatability and taste. Additionally, among VFCs, *Staphylococcus* spp. and *Macrococcus* spp. showed a strong correlation with octanal and heptanal, and meanwhile, those highly correlated with nonanal, pentanol, and oct-1-en-3-ol were *Debaryomyces* spp., *Psychrobacter* spp., and *Brochothrix* spp., respectively, providing beef with a unique flavor. *Staphylococcus* spp. was proposed to be the dominant genus for dry-aged beef. This study provides valuable reference for the understanding of the role of microorganisms involved in dry aging.

## 1. Introduction

There are two main forms of aging: wet and dry aging. Wet-aged beef is sealed in a bag using vacuum packaging technology and aged at a low residual pressure and temperature of −1 to 2 °C for 3–83 days. The beef becomes soft and juicy by virtue of its own endogenous enzymes [1]. Dry-aging technology is an aging method in which beef is aged for weeks or even months in an aging room at a temperature of 0 to 3 °C with controlled relative humidity and air-flow rate, without packaging materials, during which two distinct changes occur on the surface of the beef: the surface becomes dry due to evaporation of water, and mold and yeast grow on the surface of the beef with their distinct characteristics [2,3,4,5].

Dry-aged beef has a more desirable and distinctive flavor profile than wet-aged beef, often described as buttery, nutty, grilled, roasted nutty, brothy, and sweet [6,7]. Trained sensory evaluators concluded that wet-aged beef had a more acidic, metallic, and bloody taste, while dry-aged beef had a more meaty and grilled taste [8]. Dry-aged beef exhibits a darker hue and reduced redness in comparison to wet-aged beef. This discoloration arises from the accumulation of metmyoglobin, a product of myoglobin oxidation, and darkening due to surface dehydration. Such alterations in coloration can lead to economic losses, as consumer decisions regarding meat purchases at the retail level are predominantly influenced by visual appearance, with color being a primary factor. The key effects of dry-aging are the production of distinctive flavors and tenderization of the meat, as well as more easily digestible and absorbable nutrients, while consumers evaluate the quality of beef based on taste, aroma, juiciness, and tenderness [1]. Hence, with the development and breakthroughs of modern industrial technology in recent years, the environmental conditions required for dry-aging have become easy to achieve, and people’s living standards continue to improve; thus, dry-aging beef is becoming more and more popular [3].

During dry aging, proteins and fats are chemically broken down, resulting in a more intense nutty and beef flavor [9]. In addition, during the aging process, endogenous protein hydrolases in beef degrade myofibrillar proteins in the cytoskeleton and collagen in the intramuscular connective tissue, resulting in a more tender beef texture [10]. The improvement of beef flavor during the dry-aging process may involve the release of reducing sugars, FAAs, peptides, and the decomposition of ribonucleotides to generate IMP, GMP, inosine, and hypoxanthine [11]. These changes are related to the activity of hydrolytic enzymes; for example, calcium-dependent proteases involved in muscle structure breakage and histone proteases involved in flavor peptide production may play an important role in the flavor formation of beef [12]. During aging, endogenous enzymes in beef break down proteins into polypeptides and FAAs, of which aliphatic amino acids provide sweetness and amino acids containing sulfur atoms, such as cysteine and methionine provide umami. In addition, carbohydrates are broken down into sugars, which produce sweetness, and fats are degraded to aromatic fatty acids during aging. All of these breakdown products not only provide a rich, nutty flavor and aroma to dry-aged beef [13] but are also more easily digested and absorbed by the human body. During the aging process, interactions between flavor precursor components result in new flavor components, which contribute to further flavor enhancement [9].

With the continuous improvement of people’s economic conditions, people put forward higher requirements on the nutritional value, taste, and flavor of meat. Beef has high nutritional value, with the following nutritional components: water content of 60–70%, protein content of 20–25%, fat content of 5–15%, ash content of 1%, and carbohydrate content of less than 1%. Through the dry-aging process, the endogenous enzymes and exogenous enzymes secreted by microorganisms can decompose protein and fat into polypeptides, amino acids, and fatty acids, which are not only more easily digested and absorbed by the human body but also conducive to forming a unique flavor and tender and juicy texture, making it popular among consumers. Meanwhile, related research and industrialization of dry aging in China is currently in the initial processing stage. Therefore, through this research, consumers could be provided with more delicious beef ingredients.

To date, there are few studies on the important effects of microorganisms on beef quality indicators (e.g., taste, flavor, and texture) during aging. Therefore, it is crucial to investigate the correlation between microorganisms and the overall quality of aged beef. Therefore, the objective of this study was to analyze the correlations between quality changes and microbial revolution of dry-aged beef in order to identify the dominant genus, shorten the aging period, and increase the aging efficiency. In addition, the obtained results could provide a valuable reference for subsequent research on intensified dry-aged beef with the combination of different strains.

## 2. Materials and Methods

### 2.1. Dry-Aging Process of Beef

The beef used in this study was sourced from a local beef store (Wuxi, China); adult male cattle aged 16–20 months were slaughtered, and the beef tenderloin was then divided into chunks weighing 1.0 ± 0.2 kg. The higher humidity of 85 ± 5% was selected as the relative humidity for the dry-aged beef process, and 0.5 m/s was selected as the air-flow rate. The fresh beef was cleaned, drained, and placed in a chilled 1 °C constant temperature cabinet with the relative humidity and air-flow rate set and allowed to dry age for 21 days, with turnovers every 3 days [3]. Samples were taken at day 0, 3.5 days, 7 days, 14 days, and 21 days, respectively. All analyses were conducted at the specified aging times.

### 2.2. Determination of Textural Properties

The textural analysis was performed on cooked beef. The beef sample was cooked on a flat electric frying pan at 135 °C. When the central temperature of the sample reached 41 °C, the sample was turned over and continued frying until the central temperature reached 72 °C. The sample was cut into small pieces about 1.27 cm × 1.27 cm × 2.54 cm. Shear force: The method of Wanyu Ren et al. [14] was used as a reference. The probe was placed perpendicular to the muscle fibers of the meat sample; using a probe model HDP/KS5, a probe forward distance of 25 mm, pre-measurement speed of 5.0 mm/s, and a speed under measurement of 3.0 mm/s were used for each of the three parallel tests.

Texture Profile Analysis (TPA): The method of Zihan Zhang et al. [15] was used as a reference, with appropriate modifications. Four analysis indicators, hardness, elasticity, chewiness, and resilience were selected. A P35 probe, pre-measurement speed of 2.0 mm/s, speed under measurement of 2.0 mm/s, and compression ratio of 40% were used for each of the three parallel tests. The probe was measured twice at an interval of 5.00 s and the trigger type was automatic.

### 2.3. Determination of FAAs

The method of Szterk et al. [16] was referenced, and a high-performance liquid chromatography (HPLC) amino acid analyzer (type 1100, Agilent Technologies Co., Ltd., Santa Clara, CA, USA) was used to determine the free amino acids (FAAs) in the sample. A 1.5 g sample was accurately weighed and placed in a centrifuge tube and 15 mL 0.02 mol/L hydrochloric acid was added, homogenized for 1 min, and centrifuged at 4 °C and 5000 r/min for 10 min. Then, the supernatant was taken, the precipitation was repeated, the supernatant was combined, and the volume was fixed to 50 mL. After 2 mL of constant volume supernatant, 2 mL 8% (*v*/*v*) sulfosalicylic acid solution was added, centrifugated at 4 °C and 10,000 r/min for 10 min, and then the supernatant was filtered through a 0.22 μm filter membrane and tested by machine.

An amount of 1 nmol/μL of 17 amino acid standards (aspartate, histidine, glutamic acid, serine, glycine, threonine, alanine, arginine, tyrosine, cystine, valine, methionine, phenylalanine, isoleucine, leucine, lysine, and proline), OPA, and FMOC were purchased from Sigma. Mobile phase A (pH = 7.2): 27.6 mmo/L sodium acetate-triethylamine-tetrahydrofuran (volume ratio 500:0.11:2.5). Mobile phase B (pH = 7.2): 80.9 mmol/L sodium acetate–methanol–acetonitrile (volume ratio: 1:2:2). Agilent Hypersil ODS column (5 μm, 4.0 mm × 250 mm). The gradient elution procedure was as follows: 0 min, 8% B; 17 min, 50% B; 20.l min, 100% B; 24 min, 0% B. The flow rate of the mobile phase was 1.0 mL/min. The column temperature was 40 °C. The detection wavelength of the ultraviolet detector (VWD) was 338 nm, and the detection wavelength of the proline was 262 nm. Amino acid content was quantified with an extra-standard method. In addition, the chromatographic method lasted for 22 h. All experiments were repeated three times. 

### 2.4. Determination of Taste Activity Value (TAV)

The Taste Activity Value (TAV) was used to analyze the contribution of free amino acids (FAAs) to the overall taste of the beef during the dry-aging process [17]. The Taste Activity Value (TAV) was calculated according to Equation (1).
(1)$$TAV=\frac{C_{i}}{T_{i}}$$

Note: C_i_ is substance content, mg/100 g; T_i_ is the sensory threshold, mg/100 g.

### 2.5. Determination of VFCs

The method of Su Li [18] was referred to, and the pretreatment method of meat samples was modified appropriately. A 3.0 g sample of ground beef was accurately weighed and placed in a 25 mL sample bottle, sealed, and kept at a constant temperature in a 60 °C water bath for 15 min. SPME-GC-MS (Montauban, France) was used to determine VFCs in the dry-aged beef under different aging times. Volatile flavor compounds (VFCs) were analyzed using an internal standard method with semi-quantitative analysis, and the internal standard was 81.06 μg/mL dioctyl alcohol. Relative Odor Activity Value (ROAV) was used to analyze the contribution of volatile flavor components to the overall flavor of the sample during the dry-aging process of beef [19]. The volatile component that contributes most to the flavor of the sample (ROAVmax) is specified as 100, and other volatile flavor components were calculated using the following Equations (2) and (3):(2)$${ROAV}_{i}=\frac{C_{i}}{T_{i}}\times \frac{T_{max}}{C_{max}}\times 100$$
(3)$$OAV=\frac{C_{i}}{T_{i}}$$

Note: C_i_ is the relative content of each volatile component, %; C_max_ is the relative content of volatile components that contributes the most to the overall flavor of the sample, %; T_i_ is the sensory threshold of each volatile component, μg/kg; and T_max_ is the sensory threshold of the volatile components that contribute the most to the overall flavor of the sample, μg/kg.

### 2.6. Sensory Evaluation

Refer to GB/T 16291.1-2012 General Guidelines for Selection, Training, and Management of Evaluators for Sensory Analysis about recruiting and training the 12 sensory evaluators (male to female, 1:1). The sensory evaluation of beef fillet samples was carried out using a nine-point preference scale method. The beef sample was cooked on a flat electric frying pan at 135 °C. When the central temperature of the sample reached 41 °C, the sample was turned over and continued frying until the central temperature reached 72 °C. The sample was cut into small pieces about 1.27 cm × 1.27 cm × 2.54 cm and placed in a plastic cup marked with a pre-assigned, random, three-digit code. Five sensory attributes of color, flavor, tenderness, chewiness, and juiciness were rated with 1 being “very much dislike” and 9 being “very much like”.

### 2.7. DNA Extraction and PCR Amplification

A sterile cotton swab dipped in normal saline was used to take random samples on the surface of beef samples. The swab was put into a 2 mL centrifuge tube and 1 mL of Biechuan bacterial cracking solution was added; then, it was shaken, mixed, and cracked at 65 °C for 40 min and 70 °C for 20 min. An amount of 900 μL supernatant was taken for subsequent extraction. Microbial community genomic DNA in different samples is extracted using the OMEGA Stool DNA Kit (D4015, Omega, Inc., Santa Clara, CA, USA) according to the manufacturer’s instructions. The total DNA was eluted with 50 μL buffer in ultra-pure water and stored at −80 °C. The concentration and purity of DNA were determined using an ND1000 microspectrophotometer. The method of Sangdon et al. [5] was referenced with appropriate modifications. A forward primer 341F (5′-CCTACGGGNGGCWGCAG-3′) and reverse primer 805R (5′-GACTACHVGGGTATCTAATCC-3′) were used to amplify the highly variable region V3-V4 of bacterial 16S rDNA gene. The length of the target fragment amplified by primers 341F and 805R was about 469 bp. A forward primer fITS7 (5′-GTGARTCATCGAATCTTTG-3′) and a reverse primer ITS4 (5′-TCCTCCGCTTATTGATATGC-3′) were used to amplify the highly variable region of ITS rDNA gene ITS1-ITS2. The length of the target fragment obtained by PCR amplification was about 353 bp.

### 2.8. Illumina Sequencing and Processing of Sequencing Data

Double-end sequencing was performed using an Illumina sequencing platform 250PE according to standard operation (Hangzhou Lianchuan Biotechnology Co., Ltd., located in Hangzhou, China). The background noise was removed by DADA2 (Divisive Amplicon Denoising Algorithm) to obtain feature tables and feature sequences for further diversity analysis, species taxonomic annotation, and difference analysis. The next-generation sequencing (NGS) data for dry-aged beef samples was deposited to the NCBI BioProject database under accession number PRJNA1006360.

### 2.9. Statistical Analysis

Multivariate statistical analysis was used to study microorganisms associated with changes in FAAs, VFCs, and textural properties. Correlation analysis of quality variation with microbial diversity was performed with VIP plots, loading plots, and heat maps using SIMCA (v14.1) and TBtools software (v1.098769); plotting was performed using Origin 2022b; Partial Least Squares Regression (PLS) was performed using SIMCA 13.0 software; Spearman correlation analysis and heat map visualization of correlation data was performed using TBtools software; and SPSS 20.0 software was used for statistical analysis. All experiments were repeated three times.

## 3. Results and Discussion

### 3.1. Analysis of Textural Properties

One of the most important palatability factors affecting consumer experience is tenderness [20,21]. In the beef industry, aging is generally regarded as one of the most important factors determining the final tenderness of beef [22]. The change in tenderness can be reflected by measuring the shear force. The smaller the shear force is, the more tender the meat is [10]. As shown in our previous work [23] and Figure 1a, the shear force of the sample decreased with the extension of aging time, indicating that the tenderness of the sample gradually increased. Hanzelkova et al. [22] studied the changes in the texture characteristics of beef of different varieties during the dry-aging process and found that the shear force of beef of all varieties gradually decreased during the aging process. The main reason for this phenomenon is that endogenous proteolytic enzymes such as calpains degrade muscle fibrin and collagen in intramuscular connective tissues, while the tenderness of meat is mainly determined by two factors, myofiber and connective tissues [24]. Low levels of lipid oxidation may enable calpains to maintain longer activity [25], which corresponds to an increase in MDA content.

As can be seen from our previous work [23] and Figure 1b, the hardness, chewability, and resilience of beef samples first increased and then decreased in the dry-aging process, reaching a maximum value at 3.5 days, while the elasticity of beef tenderloin samples decreased first and then increased. The decrease in hardness and chewiness may be due to the decomposition of myofibrillar protein and collagen in connective tissue under the action of endogenous protease I, protease II, cathepsin, and enzymes secreted by microorganisms to soften muscle tissue, which is the same as the improvement mechanism of tenderness [26].

### 3.2. Analysis of FAAs

FAAs are one of the main flavor compounds in meat products, and they can also produce several VFCs through the Maillard reaction with reducing sugars [11]. As can be seen from our previous work [23], after 21 days of dry aging, the total FAA content increased from 436.31 mg/100 g to 651.92 mg/100 g, and the content of Essential Amino Acids (EAAs) increased from 124.86 mg/100 g to 156.17 mg/100 g. The content of most FAAs, such as Asp, Glu, Gly, Thr, Ala, and Pro, showed an overall increasing trend, which was consistent with Lee et al.’s findings, and this may be related to the concentration effect caused by water evaporation and the high rate of protein hydrolysis [3]. Glu is one of the most important components of meat umami, while Ala is related to sweetness. Their content increased significantly from 9.56 ± 0.70 mg/100 g and 181.17 ± 6.64 mg/100 g at day 0 to 41.18 ± 1.54 mg/100 g and 302.65 ± 4.52 mg/100 g at day 21, respectively. TAV values also increased from 0.32 and 3.02 to 1.37 and 5.04, respectively, both of which were greater than 1, indicating that they had a significant contribution to the overall taste of the sample. The greater the TAV value, the higher the contribution to the overall taste of the sample. However, the contents of a few amino acids fluctuated, such as His, Tyr, and Trp, while the contents of Cys decreased, which may be due to the occurrence of amino acid degradation reactions, such as the degradation of amino acid side chains of Tyr and Trp to produce phenols and indole [27].

### 3.3. Determention of VFCs

Aroma compounds in the dry-aging process mainly come from three sources: One is the flavor compounds of beef itself, such as aldehydes and alcohols. The other is the flavor compounds produced by lipid oxidation and degradation, such as straight-chain aldehydes, alcohols, ketones, hydrocarbons, furans, etc.; heptanal, octanal, and nonanal are the main products of aldehydes. Third, Maillard reactions between amino acids and reducing sugars produce compounds containing sulfur and nitrogen, such as pyrazine [10,28].

A total of 26 compounds were identified, as shown in our previous work [23], including six aldehydes, eight alcohols, three acids, two esters, one alkane, one alkene, one ketone, and four other compounds. The sensory threshold of different flavor compounds differs greatly. For example, the sensory threshold of acetic acid is 22,000 μg/kg, while the sensory threshold of aldehydes is relatively low, such as 1 μg/kg for nonaldehyde, so a small amount of aldehydes can also promote the flavor of meat [10].

Among the 26 VFCs, the main contribution to flavor may only stem from some of them. Therefore, the ROAV method was used in this study to evaluate the effect of flavor compounds on flavor. Compounds with ROAV ≥ 1 were the key flavor components of the sample, and compounds with ROAV < 1 ≤ 0.1 played an important role in modifying the overall flavor of the sample [19]. The greater the ROAV value, the greater the contribution to the overall flavor of the sample. As can be seen from Appendix A (see Appendix A), n-octanal, nonanal, and 1-octene-3-ol are the key flavor components in the whole dry-aging process. With the extension of aging time, n-amyl alcohol and heptanal, which have little effect on the flavor of fresh beef samples, become the key flavor components of 21-day dry-aged beef samples. Similar results were obtained by Yang et al. [29]. In addition, 3-hydroxy-2-butanone, vinyl caproate, and 2-n-amylfuran are the key flavor components in the aging process. Nonaldehyde and n-amyl alcohol contributed the most to the flavor of samples at different aging time periods. Nonaldehyde had floral, citrus, grass, and other aromas, while n-amyl alcohol had fruit and spices. These results were consistent with the study by Lee et al. [3].

### 3.4. Sensory Evaluation of Dry-Aged Beef

The sensory evaluation results of beef samples during dry aging are shown in Table 1. The score of the flavor preference index was 5.48–7.20, the tenderness preference index was 5.40–7.01, the chewability preference index was 6.04–7.23, the juiciness preference index was 5.81–6.47, and the color preference index was 6.26–6.92. The scores of flavor, tenderness, chewability, and juiciness of samples at 21 days were significantly increased compared with those at 0 d (*p* < 0.05), and the color score was the highest at 7 days. Campbell et al. [6] found that the juiciness and tenderness of beef aged 16–21 days after dry aging were higher than that of raw beef, which was consistent with the results of this study.

### 3.5. High-Throughput Sequencing

The basic information of bacterial and fungal sequencing of samples with different aging times of the dry-aged beef was shown in Appendix A (see Appendix A). After optimization, a total of 955,073 valid bacterial 16S rDNA sequences with an effective sequence length of 404.51 M and 1,191,497 valid fungal ITS rDNA sequences with an effective sequence length of 325.09 M were obtained, respectively.

### 3.6. Alpha Diversity and Feature Distribution of Bacterial Community

As shown in Appendix A (see Appendix A), the Goods_coverage of all five samples was 1, indicating that the sequencing results accurately reflected the real situation of bacteria contained in each sample. The changes in the values of Shannon, Simpson, and Chao1 showed that the abundance and diversity of bacteria in the samples first decreased and then fluctuated as aging proceeded, decreasing to 2.64, 0.65, and 37.33 at 21 days. This might be related to the gradual disappearance of bacteria that could not grow and reproduce at low temperatures and the gradual formation of dominant species during aging.

The Venn diagram could visually indicate the number of features common and unique to each sample, as shown in Figure 2. The number of features in the dry-aged beef samples gradually decreased as the aging time increased to only 52 features at 21 days, which was consistent with the change in bacterial α-diversity. During the dry-aging process, the sample-specific features were 415, 140, 12, 11, and 9, respectively, and from day 0 to 3.5 days, the unique features decreased by 66.3%. At 21 days, the total number of features and the unique features were the lowest of all samples, indicating that a large proportion of bacterial growth was inhibited at this time, which may be related to the growth of the dominant species.

### 3.7. Taxonomic Composition of Bacterial Community

After comparison with the Sliva database, there were 22 phyla of bacterial communities in the dry-aged beef process, as shown in Figure 3a, among which, the four phyla with high relative abundance were mainly Proteobacteria, Actinobacteria, Firmicutes, and Bacteroidetes. The relative abundance of Proteobacteria gradually increased and reached the highest level of 97.73% at 14 days. The relative abundance of Actinobacteria gradually decreased from 28.10% to 1.28%, the relative abundance of Firmicutes fluctuated, and the relative abundance of Bacteroidetes gradually decreased to 0 at 7 days. Proteobacteria were the dominant phylum in the whole dry-aging process, which was the largest phylum among bacteria with a wide variety of species and metabolic forms.

The species and relative abundance of bacteria varied greatly at different aging times. The bacterial features of each sample of dry-aged beef were analyzed at the genus level, as shown in Figure 3b. The dominant bacterial genus detected in the fresh beef was Acinetobacter, but its relative abundance generally tended to decrease with increasing aging time and was only 2.50% at 21 days. The dominant bacterial genus during aging was Psychrobacter, usually Gram-negative aerobic cocci, whose relative abundance increased and then decreased, reaching a maximum of 73.40% at 21 days. The relative abundance of Pseudomonas suddenly increased to 31.30% at 14 days but decreased to 1.96% at 21 days. The relative abundance of Staphylococcus fluctuated and reached the highest relative abundance of 4.12% at 3.5 days. It had strong protease and lipase activities, which improved the texture and flavor of fermented meat products.

In order to study the similarity and difference of samples at different aging time points during the dry-aging process, hierarchical clustering was performed at the genus level and a distant clustering tree structure was constructed to directly observe the similarity between specific samples, with shorter branches representing a more similar sample species composition. In Figure 4, it can be seen that the changes in bacterial colonies of the dry-aged beef samples did not show regular changes at different aging time points, and the bacterial communities of different parallel samples taken at the same time varied greatly, such as the parallel samples at 3.5 days, 7 days, and 14 days belonged to different branches. This might be related to the uneven growth of microorganisms caused by dry aging under natural conditions of maturation.

### 3.8. Alpha Diversity and Feature Distribution of Fungal Community

As shown in Appendix A (see Appendix A), the Goods_coverage values of fungal diversity of the dry-aged beef samples were all 1, indicating that the results of this sequencing can truly reflect the microorganisms in the samples. The Chao1 index decreased during the dry-aging process, indicating that the fungal richness in the samples decreased. The Shannon and Simpson indices showed an overall decreasing trend, with the lowest value at 21 days, indicating that the diversity, richness, and uniformity of fungi decreased, but the Shannon and Simpson indices were higher at 14 days than at 7 days and 21 days, indicating that the diversity and richness of fungi at 14 days were higher. The Shannon index and Simpson index were higher at 14 days than at 7 days and 21 days, indicating that the diversity, richness, and evenness of fungi increased at 14 days. When analyzed in combination with bacterial diversity, the decrease in relative abundance of Psychrobacter at 14 days may be related to the increase in fungal diversity and abundance.

As shown in Figure 5, there were 128 fungal features in fresh beef and 76 features at 21 days of dry-aged beef, with an overall decreasing trend, reaching a minimum value of 60 at 14 days. The number of endemic features at each time point was 60, 37, 22, 16, and 29, respectively, and the 14-day samples had the lowest number of endemic features and total number of features. This contradicts the changes in the Shannon and Simpson indices, which may be due to the fact that Shannon and Simpson indices characterize diversity and evenness together, while the number of features indicates the actual number of species observed, reflecting the richness of the flora in the sample without considering the evenness of each species in the bacterial community.

### 3.9. Taxonomic Composition of Fungal Community

Comparative analysis with the Unite database showed that there were five phyla of fungi at five time points during the dry-aging process, as shown in Figure 6a. Ascomycota, Basidiomycota, Zygomycota, and Mortierellomycota. Ascomycota and Basidiomycota were predominant in fresh beef with a total relative abundance of 99.31%, and Ascomycota was predominant after 21 days of dry-aging with a relative abundance of 99.88%. During the dry-aging process, the Ascomycota was the dominant phylum, increasing to 98.90% at 7 days. Since then, the relative abundance was greater than 99%. The relative abundance of the Basidiomycota phylum reached a maximum of 14.97% at 3.5 days and decreased rapidly thereafter. The relative abundance of Zygomycota phylum decreased gradually and reached the highest value of 0.69% at day 0. The relative abundance of Mortierellomycota was 0 at the first 14 days and 0.002% at 21 days.

The fungal features of the samples at different aging time points during the dry-aging of beef were classified at the genus level, and the results are shown in Figure 6b. A total of seven fungal genera with a relative abundance above 1% were identified during the dry-aging process, and the dominant genus was Candida, whose relative abundance increased from 67.10% at day 0 to 98.49% at 21 days. The relative abundance of Petromyces, Trichosporon, and Malassezia showed an overall significant decrease with maturation. The relative abundance of Debaryomyces fluctuated and was 1.09% in 21-day samples.

### 3.10. Correlation between Microbial Changes and Texture

According to Figure 7a, the results of strain Variable Importance Project (VIP) analysis related to texture characteristics showed that the importance index of independent variables (VIP (pred)) of 18 genera varied from 0.4411 to 1.5656, and there were seven genera with VIP (pred) > 1.0, among which VIP (pred) > 1.5 were *Staphylococcus* spp. and *Macrococcus* spp. The correlation between microbe and textural properties at different aging times was analyzed. As shown in Figure 7b, R2x(cum) was 0.805, R2y(cum) was 0.642, and Q2(cum) was 0.18. *Staphylococcus* spp. and *Macrococcus* spp. have a strong correlation with chewability and hardness. The tenderness, elasticity, and resilience did not show any correlation with microorganisms.

### 3.11. Correlation between Microbial Changes and FAAs

VIP values can directly reflect the overall contribution of variables to model classification. The VIP analysis results related to FAAs showed that VIP (pred) of 18 genera varied from 0.60473 to 1.2953, among which 11 genera had VIP (pred) > 1.0. As shown in Figure 8a, the most important bacteria genera to FAAs are *Staphylococcus* spp., *Macrococcus* spp., *Serratia* spp., *Carnobacterium* spp., *Corynebacterium* spp., and *Kocuria* spp. The most important fungal genera are *Malassezia* spp., *Trichosporon* spp., *Petromyces* spp., *Debaryomyces* spp., and *Candida* spp. The VIP (pred) values of these bacteria were all above 1.0, indicating that these bacteria may play a leading role in the production of FAAs during the dry-aging process of beef.

In this study, Partial Least Squares Regression (PLS) was used to analyze the correlation between FAAs and microbial changes at different aging times. The model was constructed with bacteria and fungi with relative abundance greater than 0.1% at the genus level as the independent variable X, 18 species of FAAs as the dependent variable Y, and different aging times as the observed variables. The cumulative statistic R2X (cum) of the prediction component of the fitting model was 0.818, the model interpretation rate parameter R2Y (cum) was 0.842, and the prediction ability parameter Q2 (cum) was 0.572, indicating that the model had good prediction ability. The correlation between X and Y can be judged according to the distance between them.

As can be seen from Figure 8b, the correlations between microbial changes and FAAs are complicated. A single free amino acid can be associated with a variety of microorganisms, and a single microorganism can also be associated with a variety of FAAs. *Staphylococcus* spp. and *Macrococcus* spp. were highly correlated with Ala. *Serratia* spp. was highly correlated with Phe, Val, Lys, Pro, and Ser. *Psychrobacter* spp. was strongly related to Glu. Among the fungi, *Candida* spp. had a strong correlation with Thr, Gly, and Glu. Cys was strongly associated with *Acinetobacter* spp. in bacteria and *Malassezia* spp. and *Petromyces* spp. in fungi, while *Carnobacterium* spp., *Brochothrix* spp., and *Pantoea* spp. showed no correlation with FAAs.

### 3.12. Correlation between Microbial Changes and VFCs

As can be seen from Figure 9a, the results of strain VIP analysis related to VFCs showed that VIP (pred) of 18 genera varied from 0.55232 to 1.1984, and there were nine genera with VIP (pred) >1.0: in order, *Staphylococcus* spp., *Macrococcus* spp., *Brochothrix* spp., *Kocuria* spp., *Acinetobacter* spp., *Enhydrobacter* spp., *Psychrobacter* spp., *Candida* spp., and *Debaryomyces* spp. The correlation between microbial changes and VFCs at different aging times was analyzed. The results are shown in Figure 9b, R2x(cum) is 0.825, R2y(cum) is 0.775, and Q2(cum) is 0.502, indicating that this model has good predictive ability. The correlations between microorganisms and VFCs were diverse. *Staphylococcus* spp. and *Macrococcus* spp. are highly correlated with 2-pentylfuran, heptanal, and octanal. *Brochothrix* spp. was highly correlated with 1-octen-3-ol. *Kocuria* spp., *Enhydrobacter* spp., and *Debaryomyces* spp. were strongly correlated with nonanal and furfural. *Psychrobacter* spp. was strongly related to 1-pentanol. *Aeromonas* spp. had a strong correlation with 3-hydroxy-2-butanone. *Serratia* spp. and *Malassezia* spp. showed no correlation with volatile flavorings. Overall, bacteria showed a stronger correlation to volatile flavors than fungi.

In the dry-aging process of beef, the three FAAs, Ala, His, and Glu, made significant contributions to the overall taste of the samples, especially the sweet amino acid Ala, whose TAV value was 5.04, which was much higher than the other two. The higher the TAV value, the higher the contribution to the overall taste of the samples. *Staphylococcus* spp. and *Macrococcus* spp. were highly correlated with Ala. Among the VFCs, nonanal, pentanol, octanal, oct-1-en-3-ol, and heptanal are the key flavor compounds in the aging process. *Kocuria* spp., *Enhydrobacter* spp., and *Debaryomyces* spp. were highly correlated with nonanal, while *Psychrobacter* spp. was highly correlated with pentanol, and *Staphylococcus* spp. and *Macrococcus* spp. were highly correlated with octanal and heptanal. *Brochothrix* spp. had a strong correlation with oct-1-en-3-ol, and *Brochothrix* spp. is a meat spoilage bacterium [30]. In terms of textural properties, *Staphylococcus* spp. and *Macrococcus* spp. were strongly correlated with chewability, hardness, and palatability of beef. It was found that *Staphylococcus* spp. and *Macrococcus* spp. were strongly related to the formation of flavor and aroma during the dry-aging process of beef. Referring to the existing literature, *Staphylococcus* spp. is a commonly used strain for meat products with lipase and protease activity and is used in fermentation to produce unique flavor compounds, delay fat oxidation, and stabilize color [31], including *Staphylococcus xylosus* [32], *Staphylococcus sarcosa* [10], etc., whereas *Macrococcus* spp. caused a rotten egg taste and turbidity in beer. Hence, *Staphylococcus* spp. was identified as the dominant acting genera in the dry-aging process of beef, and strains with good fermentation and enzymatic properties were screened.

## 4. Conclusions

In conclusion, by monitoring the quality changes and microbiota revolution of beef during the dry-aging process and analyzing the correlation between quality changes and microbiota revolution, it could be concluded that after 21 days of dry aging, in terms of textural properties, *Staphylococcus* spp. and *Macrococcus* spp. were strongly correlated with chewiness and hardness and provided beef with unique palatability. Three FAAs (e.g., Ala, His, and Glu) contributed significantly to the overall taste of dry-aged beef, especially the sweet amino acid (Ala), and the strains highly correlated with Ala were *Staphylococcus* spp. and *Macrococcus* spp. Nonanal, pentanol, octanal, oct-1-en-3-ol, and heptanal were confirmed to be key flavor compounds in the aging process. *Kocuria* spp., *Enhydrobacter* spp., and *Debaryomyces* spp. were highly correlated with nonanal, while *Psychrobacter* spp. showed a strong correlation with pentanol. *Staphylococcus* spp. and *Macrococcus* spp. are likely to be highly correlated with octanal and heptanal, as is the case with *Brochothrix* spp. and oct-1-en-3-ol. However, the dry-aging process still maintains problems such as long cycle time, high cost, unstable quality, high loss, lack of standardization, and safety of contamination by trash bacteria. There is an urgent need to develop proper strains for intensified dry aging. This work revealed the correlation between microbiota and qualities of dry-aged beef at a microscopic level, providing a guide for subsequent research on both mechanisms and practical manufacturing of dry-aged beef.

## Figures and Tables

**Figure 1 foods-13-01552-f001:**
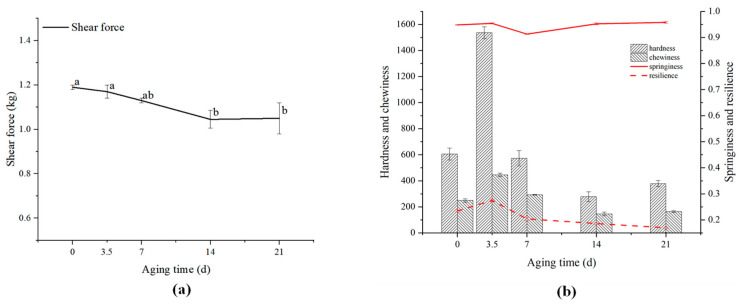
Changes of texture characteristics of the dry-aged beef. (**a**) Shear force. (**b**) TPA. Different lowercase letters indicate significant differences (*p* < 0.05).

**Figure 2 foods-13-01552-f002:**
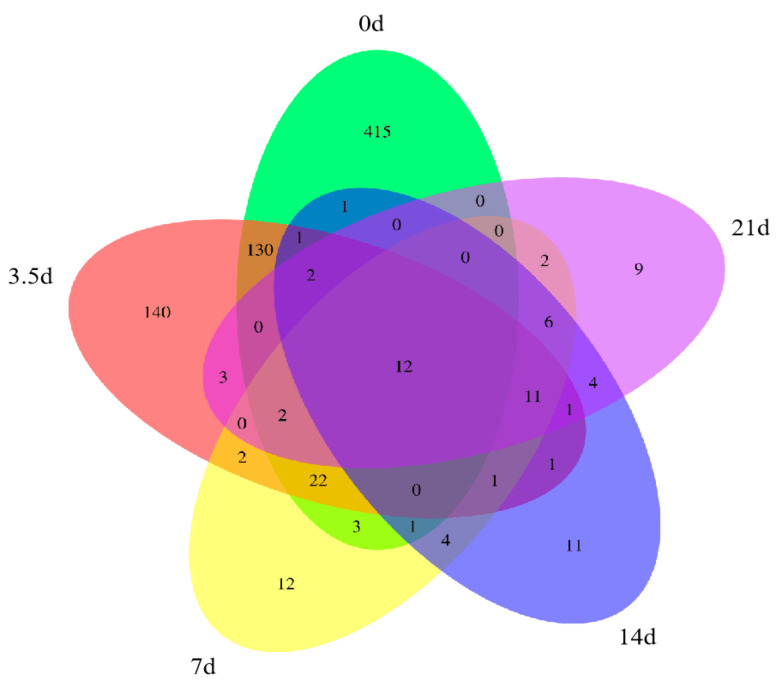
Bacteria Venn diagram of dry-aged beef.

**Figure 3 foods-13-01552-f003:**
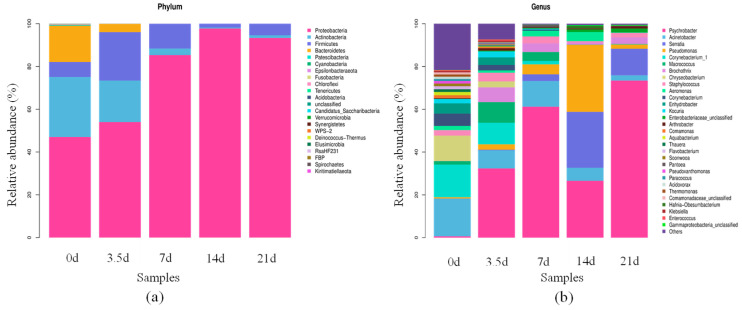
Bacterial distribution at phylum and genus level of beef at different aging times. (**a**) Phylum taxonomic level. (**b**) Generic taxonomic level.

**Figure 4 foods-13-01552-f004:**
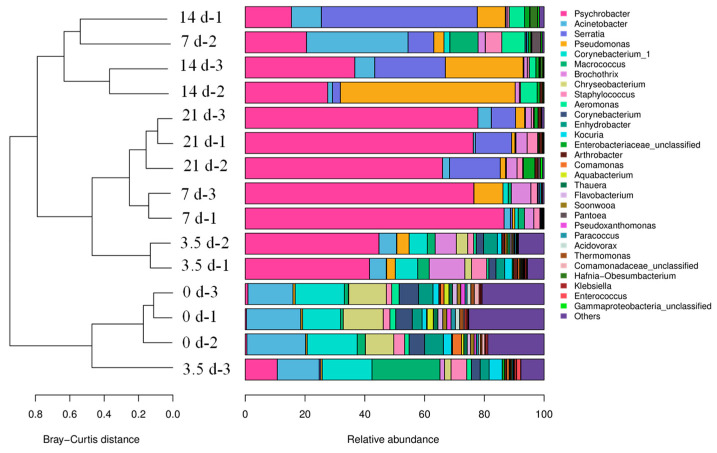
Hierarchical cluster analysis of beef samples at different times of dry aging.

**Figure 5 foods-13-01552-f005:**
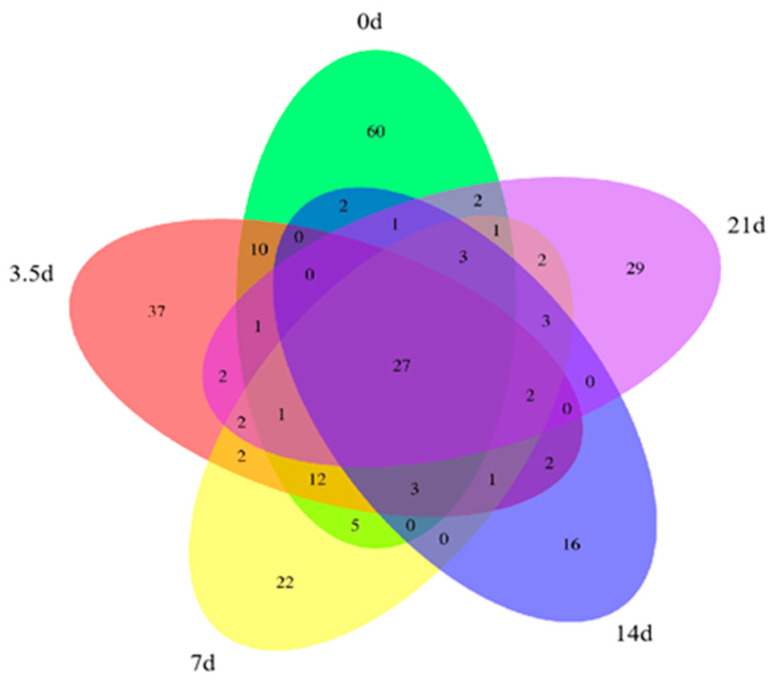
Fungi Venn diagram of dry-aged beef.

**Figure 6 foods-13-01552-f006:**
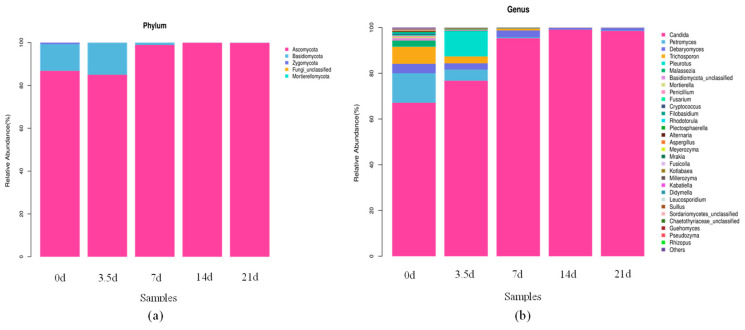
Fungi distribution at phylum and genus levels of beef at different aging times. (**a**) Phylum taxonomic level. (**b**) Generic taxonomic level.

**Figure 7 foods-13-01552-f007:**
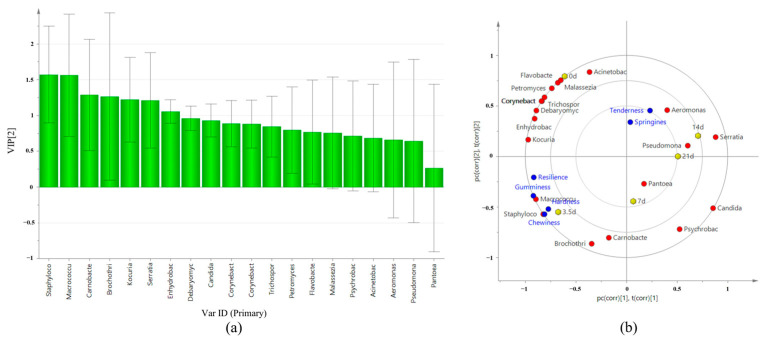
VIP map and correlation analysis of texture and major microorganisms. (**a**) VIP diagram. (**b**) Correlation load diagram.

**Figure 8 foods-13-01552-f008:**
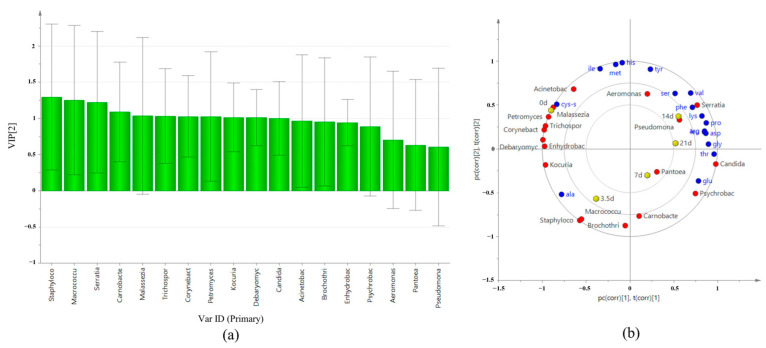
VIP map and correlation analysis of FAAs and major microorganisms. (**a**) VIP diagram. (**b**) Correlation load diagram.

**Figure 9 foods-13-01552-f009:**
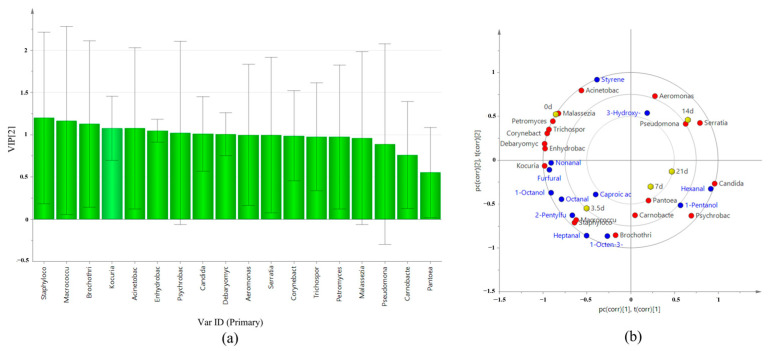
VIP map and correlation analysis of VFCs and major microorganisms. (**a**) VIP diagram. (**b**) Correlation load diagram.

**Table 1 foods-13-01552-t001:** Sensory evaluation of dry-aged beef.

Group	Flavor	Tenderness	Chewiness	Succulence	Color and Luster
Day 0	5.92 ± 0.02 ^d^	5.61 ± 0.33 ^d^	6.61 ± 0.13 ^b^	6.33 ± 0.03 ^b^	6.72 ± 0.07 ^b^
3.5 days	5.48 ± 0.08 ^e^	5.40 ± 0.21 ^d^	6.04 ± 0.04 ^c^	5.93 ± 0.03 ^d^	6.26 ± 0.77 ^c^
7 days	6.93 ± 0.03 ^b^	6.21 ± 0.45 ^c^	7.07 ± 0.85 ^a^	6.08 ± 0.18 ^c^	6.92 ± 0.02 ^a^
14 days	6.53 ± 0.15 ^c^	6.67 ± 0.68 ^b^	7.01 ± 0.52 ^a^	5.81 ± 0.49 ^d^	6.67 ± 0.07 ^b^
21 days	7.20 ± 0.10 ^a^	7.01 ± 0.40 ^a^	7.23 ± 0.23 ^a^	6.47 ± 0.07 ^a^	6.71 ± 0.11 ^b^

Different lowercase letters indicate significant differences (*p* < 0.05).

## Data Availability

The original contributions presented in the study are included in the article/Appendix A, further inquiries can be directed to the corresponding author. NGS data for dry-aged beef samples were deposited to the NCBI BioProject database under accession number PRJNA1006360.

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
