# Peer review of "Study on the Correlations between Quality Indicators of Dry-Aged Beef and Microbial Succession during Fermentation"

_foods, 2024, doi:10.3390/foods13101552_

Round 1

Reviewer 1 Report

Comments and Suggestions for Authors

The current manuscript reports on the correlation of volatile compounds and free amino acids of dry-aged beef during ferementation with specific microorganisms (bacteria and fungi). The metbolites of microorganisms during the aging of foods are very important to be monitored for the food indutry, as some may be well exploited. In this context the study has a novel output. Regarding the techniques used, in my opinion, are enough given the numerous data that were obtained. Figures need to be modified as some are unclear. In addition, the authors must provide microorganisms with italics when refferring to species.

Regarding these correlations can the authors propose a mechanism in a form of graphical abstract? This would benefit better the quality of the study.

Finally, given the originality of this work I advise the authors to reduce the similatrity level of the article below 20%, concerning the ithenticate metrics.

As a conclusion, I reccomend a revision (minor) prior to further consideration for publication. My comments are within the attached pdf.

Comments on the Quality of English Language

The English language is in general good. Moderate changes are required. My suggestions are within the attached pdf.

Reviewer 2 Report

Comments and Suggestions for Authors

In the study, the researchers analyzed the relationships between dry-aged beef's quality indicators and microbial succession. This work is significant because it provides insights into how microbial communities influence the flavor and texture of dry-aged beef, a topic that has not been extensively researched before. A weakness of the study is the absence of sensory analysis, and the authors did not report whether the study was replicated or the number of repetitions for each analysis. Further detailed comments are provided below.

Lines 89 – 96: The description of the dry aging process should be more detailed. Information such as the specific cut of beef used, its size, and weight should be included to understand the impact on the aging process better. Additionally, the reasoning behind the chosen aging period should be clarified with references to existing literature on dry aging processes.

Line 95: The choice of the aging period should be justified with scientific references or previous studies that have utilized similar time frames, providing a rationale for why this specific duration was effective or chosen.

Lines 95 – 96: It should be clarified whether all analyses were conducted at the specified aging times to ensure clarity on the timeline and consistency of the experimental procedures.

Lines 98 – 105: The method section regarding textural analysis should be written in a continuous narrative, detailing whether the analyses were performed on raw or cooked meat. The specifics of the analyses, including the type of probe used, units of measurement, and the number of compression cycles, should be better detailed.

Line 106 and 114: All acronyms should be written out in full to ensure clarity and ease of understanding for all readers, specifying the number of repetitions for these analyses.

Lines 110 – 111: The method for determining the sensory threshold should be explained. It should also be mentioned why sensory analysis was not performed, as this is a significant limitation in studies involving food texture and flavor.

Lines 114-126: The analysis technique for volatile compounds should be more descriptive. It should be clarified if a quantitative technique was used and the reasons behind the choice of technique. The method used for calculating Relative Odor Activity Values (ROAVs) should be described in greater detail in the manuscript. It's important for the authors to explain their methodology more thoroughly, including any assumptions made and the specific calculations involved.

Why results are presented only up to the 21st day instead of 28 days?

Table 1 Inquiry: The area of volatile compounds listed should be clarified to better understand their concentrations and impact on the overall flavor profile of the aged beef.

Reviewer 3 Report

Comments and Suggestions for Authors

General comments

The manuscript Study on the correlations between quality indicators of dry aged beef and microbial succession during fermentation, proposed for publication in Foods aimed to monitor the quality changes of beef and microbial revolution during dry aging process, and further to analyze their potential correlation in order to identify and select the dominant acting genus.

Introduction

- I suggest adding a paragraph in which you discuss the lipid and protein oxidation process that occurs in extended aging periods. 

Dry-aged beef exhibits a darker hue and reduced redness in comparison to wet-aged beef. This discoloration arises from the accumulation of metmyoglobin, a product of myoglobin oxidation, and darkening due to surface dehydration. Such alterations in coloration can lead to economic losses, as consumer decisions regarding meat purchases at the retail level are predominantly influenced by visual appearance, with color being a primary factor.

- L 76-87: The aim of the study is not clearly formulated. I suggest that you clearly state the specific research problem that the study aims to address.

Materials and Methods

- L 107-109: please describe the method for sample preparation in order to chromatographic analyze the meat samples.

- L 109: add the specific details for HPLC analysis: chromatographic column with specifications, mobile phase, gradient or isocratic conditions, flow, column temperature, wavelength, derivatization, analytical standards etc.

- Add a chromatogram of the free amino acids in meat samples.

-  How long does the chromatographic method last (for free amino acid analysis)?

- The Taste Activity Value (TAV) should be separated from the free amino acids’ determination since is not an equation for amino acid determination.

- L 118: mention the equipment (+ city, country) used for Relative Odor Activity Value (ROAV) analysis

Results and discussion

- L 184: The clarity of the images needs enhancement. Please consider improving them for better visibility and understanding.

-  I recommend moving the free amino acids profile table from supplementary material to the manuscript since it provides essential information regarding the quality of dry-aged beef studied in the present paper.

- L 188-190: total FAAs content and EAAs content are not presented in Table S1. Please include it.

- L 200: did you consider the protein oxidation processes (oxidation of thiol groups)?

- L 186-202: Please discuss the obtained results in relation to the previously published studies.

- L 211: “SPME-GC-MS was used to determine VFCs in the dry-aged beef” – this information should be added to the Materials and Methods section.

- In Table S2 are presented results from the present study? You added to citations in the title of Table S2.

- L 231: the same question for Table 1 of the manuscript.

- L 204-230: the results are not discussed. You should not only summarize the findings but also provide a nuanced analysis of their implications, limitations, and future directions for research.

- The authors have not interpreted their findings in light of previous research. The paper lacks a discussion of how the results align with or diverge from existing scientific literature. I consider that this is a superficial understanding of the topic and a failure to critically analyze the data.

- I suggest adding a clear delineation of future research directions and limitations of the study.

Round 2

Reviewer 2 Report

Comments and Suggestions for Authors

The authors made the modifications suggested, and the manuscript quality improved significantly.

Reviewer 3 Report

Comments and Suggestions for Authors

The authors have adequately addressed all the comments and suggestions in the revised version of the manuscript.

The manuscript has been carefully revised and improved.